# Characterization of a Novel Endophytic Actinomycete, *Streptomyces physcomitrii* sp. nov., and Its Biocontrol Potential Against *Ralstonia solanacearum* on Tomato

**DOI:** 10.3390/microorganisms8122025

**Published:** 2020-12-18

**Authors:** Xiaoxin Zhuang, Congting Gao, Chenghui Peng, Zhiyan Wang, Junwei Zhao, Yue Shen, Chongxi Liu

**Affiliations:** 1Key Laboratory of Agricultural Microbiology of Heilongjiang Province, Northeast Agricultural University, Harbin 150030, China; zxx221661@163.com (X.Z.); g2456692645@163.com (C.G.); ppoapeng@163.com (C.P.); zhaojunwei@neau.edu.cn (J.Z.); 2State Key Laboratory of Phytochemistry and Plant Resources in West China, Kunming Institute of Botany, Chinese Academy of Sciences, Kunming 650201, China; zhiyan_w@163.com

**Keywords:** *Ralstonia solanacearum*, antagonistic activity, biocontrol, *Streptomyces physcomitrii* sp. nov., moss

## Abstract

Bacterial wilt of tomato is a destructive disease caused by *Ralstonia solanacearum* throughout the world. An endophytic actinomycete with antagonistic activity, designated strain LD120^T^, was isolated from moss (*Physcomitrium sphaericum* (Ludw) Fuernr). The biocontrol test demonstrated that co-inoculation by the isolate and the pathogen gave the greatest biocontrol efficiency of 63.6%. Strain LD120^T^ had morphological characteristics and chemotaxonomic properties identical to those of members of the genus *Streptomyces*. The diamino acid present in the cell wall was LL-diaminopimelic acid. Arabinose, glucose, rhamnose, and ribose occured in whole cell hydrolysates. The menaquinones detected were MK-9(H_4_), MK-9(H_6_), MK-9(H_8_), and MK-9(H_2_). The polar lipid profile was found to contain diphosphatidylglycerol, phosphatidylethanolamine, and phosphatidylinositol. The major cellular fatty acids were found to be iso-C_16:0_, iso-C_17:0_, anteiso-C_15:0_, and C_16:1_ ω7c. The DNA G+C content of the draft genome sequence, consisting of 7.6 Mbp, was 73.1%. Analysis of the 16S rRNA gene sequence showed that strain LD120^T^ belongs to the genus *Streptomyces*, with the highest sequence similarity to *Streptomyces azureus* NRRL B-2655^T^ (98.97%), but phylogenetically clustered with *Streptomyces anandii* NRRL B-3590^T^ (98.62%). Multilocus sequence analysis based on five other house-keeping genes (*atpD*, *gyrB*, *rpoB*, *recA*, and *trpB)* and the low level of DNA–DNA relatedness, as well as phenotypic differences, allowed strain LD120^T^ to be differentiated from its closely related strains. Therefore, the strain was concluded to represent a novel species of the genus *Streptomyces*, for which the name *Streptomyces*
*physcomitrii* sp. nov. was proposed. The type strain was LD120^T^ (=CCTCC AA 2018049^T^ = DSM 110638^T^).

## 1. Introduction

*R. solanacearum*, a soil-borne phytopathogen that causes vascular wilt disease, is considered the second most destructive plant pathogenic bacteria [1]. The bacterial wilt disease was first observed in southern potato tubers in the United States [2]. In 1896, its causal agent was described [3]. The pathogen can infect more than 450 plant species belonging to 54 families and is particularly devastating to tomato, leading to huge agricultural losses in tropical, subtropical, and warm-temperature regions of the world [4,5]. Up till now, no effective chemical management strategy for this disease has been available [6]. Therefore, new natural resources or antibiotics for controlling this disease are urgent.

The use of antagonistic bacteria as biocontrol agents is a promising approach for the management of soil borne pathogens. Some species have been commercialized for biological control, such as *Bacillus subtilis* (QST^®^), *Pseudomonas syringae* (Bio-save^®^), *Metschnikowia fructicola* (Shemer^®^), and *Streptomyces lydicus* WYEC108 (Actinovate*^®^*) [7]. Recently, some studies showed that biological control of bacterial wilt disease using antagonistic bacteria could be achieved [8,9,10]. Antagonistic bacteria is also proved to be effective in controlling bacterial wilt disease under field conditions [11].

Endophytic actinobacteria are those that inhabit various tissues or organs of healthy plants at specific growth stages or whole stages of their life cycle without harm to host plants [12]. Many members of this taxonomic group contribute to plant natural defenses through the production of antibiotics or induction of systemic disease resistance [13,14,15]. It has been also reported that several endophytic actinobacteria have the potential as biocontrol agents against economically important plant pathogens, such as *Botryosphaeria dothidea* [16], *Fusarium oxysporum* [15], and *Sclerotinia sclerotiorum* [17]. Hence, we have reason to believe that endophytic actinobacteria are resources for controlling bacterial wilt disease.

During our search for endophytic actinobacteria with biocontrol potential against *R. solanacearum*, an antagonistic actinobacteria, strain LD120^T^, was isolated from moss (*P. sphaericum* (Ludw) Fuernr). In this study, we performed a polyphasic taxonomic analysis on this strain and proposed that strain LD120^T^ represents a new species of the genus *Streptomyces*. The in vivo bioassays demonstrated that the strain had good biocontrol potential against bacterial wilt disease on tomato.

## 2. Materials and Methods

### 2.1. Strains

Strain LD120^T^ was isolated from moss (*P. sphaericum* (Ludw) Fuernr) collected in Kunming, Yunnan Province, southwest China (25°21′ N, 102° 92′ E). The moss sample was air-dried for 24 h at room temperature and then washed in sterile distilled water with an ultrasonic step (160 W, 15 min) to remove the surface soil completely. After drying, the sample was subjected to a seven-step surface sterilization procedure, as described by Liu et al. [14]. After being thoroughly dried under sterile conditions, the sample was ground with a mortar and pestle, employing 1 mL of 0.5 M potassium phosphate buffer (pH 7.0) per 100 mg tissue. Tissue particles were allowed to settle down at 4 °C for 20–30 min, and the supernatant was spread on cellulose-proline agar (CPA) [18] supplemented with cycloheximide (50 mg/L) and nalidixic acid (20 mg/L). After 28 days of aerobic incubation at 28 °C, the single colony was transferred and purified on oatmeal agar (International *Streptomyces* Project (ISP) medium 3) [19] and maintained as glycerol suspensions (20%, *v*/*v*) at −80 °C. The reference strains, *Streptomyces azureus* NRRL B-2655^T^ and *Streptomyces anandii* NRRL B-3590^T^, were purchased from China General Microbiological Culture Collection Center (CGMCC) for comparative analysis.

### 2.2. Phenotypic Characterization

Spore morphology was observed using scanning electron microscopy (Hitachi SU8010, Hitachi Co., Tokyo, Japan) after cultivation on ISP3 medium at 28 °C for 28 days. Samples for scanning electron microscopy were prepared, as described by Jin et al. [20]. Cultural characteristics were determined after 14 days at 28 °C using ISP media 1–7 [19], modified Bennett’s agar (MBA) [21], nutrient agar (NA) [22], and Czapek’s agar (CA) [22]. Growth at different temperatures (4, 10, 15, 20, 28, 35, 37, 40, and 45 °C) was determined on ISP3 medium after incubation for 14 days. Tolerance of pH range (pH 4–12, at intervals of 1 pH units), using a buffer system, was described by Zhao et al. [23], and NaCl tolerance (0–12%, with an interval of 1%, *w*/*v*) for growth was determined after 3 days of growth in TSB broth [24] in shake flasks (250 rpm) at 28 °C. The utilization of sole carbon and nitrogen sources was carried out, as described previously [25]. Other physiological and biochemical characteristics were conducted, according to a previous report [26].

### 2.3. Chemotaxonomic Characterization

Biomass for chemical studies was prepared by growing strain LD120^T^ in GY broth [27] in shake flasks at 28 °C for 3 days. The isomers of diaminopimelic acid (DAP) in the cell wall hydrolysates were derivatized according to the method of McKerrow et al. [28] and analyzed by HPLC using an Agilent TC-C18 Column (250 × 4.6 mm i.d. 5 µm) [26]. The whole-cell sugars were analyzed by thin-layer chromatography (TLC, Qingdao Marine Chemical Inc., Qingdao, China) [29]. Phospholipids in cells were examined by two-dimensional TLC and identified according to the procedures developed by Minnikin et al. [30]. Menaquinones were extracted from freeze-dried biomass and purified using the method of Collins [31]. The extracts were analyzed by a HPLC-UV method, as described previously [26]. Fatty acids were prepared and analyzed by GC-MS according to the method of Zhuang et al. [32].

### 2.4. Phylogenetic Analysis

Strain LD120^T^ was cultured in GY broth for 3 days at 28 °C to harvest cells. The genomic DNA was isolated using a Bacteria DNA Kit (TIANGEN Biotech, Co. Ltd., Beijing, China). The primers and procedure for PCR amplification were performed as described by Wang et al. [33]. The purified PCR product was cloned into the vector pMD19-T (Takara, Shiga, Japan) and sequenced by using an Applied Biosystems DNA sequencer (model 3730XL, Applied Biosystems Inc., Foster City, CA, USA). The almost complete 16S rRNA gene sequence (1522 bp) was uploaded to the EzBioCloud server (Available online: https://www.ezbiocloud.net/) [34] to calculate pairwise 16S rRNA gene sequence similarity between strain LD120^T^ and related similar species. Phylogenetic trees were constructed using the neighbor-joining [35] and maximum likelihood [36] algorithms using MEGA 7.0 software [37]. The confidence values of branches were assessed using bootstrap resampling with 1000 replication [38]. A distance matrix was calculated using Kimura’s two-parameter model [39]. All positions containing gaps and missing data were eliminated from the dataset (complete deletion option). Phylogenetic relationships of strain LD120^T^ were also confirmed using sequences for five concatenated housekeeping genes (*recA*, *gyrB*, *atpD*, *rpoB*, and *trpB*). The sequences of LD120^T^ were obtained from the whole genome. The sequences of each locus were aligned using MEGA 7.0 software and trimmed manually at the same position before being used for further analysis. Phylogenetic analysis was performed as described above.

### 2.5. Genome Analysis

Genomic DNA was extracted using the lysozyme-sodium dodecyl sulfate-phenol/chloroform method [40]. The whole genome was sequenced on the HiSeq 2500 Sequencing System (Illumina, San Diego, CA, USA), according to the user guide, and assembled on MiSeq plateform [41]. The digital DNA-DNA hybridization (dDDH) and average nucleotide identity (ANI) values were determined between the draft genome sequences of strain LD120^T^ and *S. azureus* NRRL B-2655^T^ and *S. anandii* NRRL B-3590^T^ online at http://ggdc.dsmz.de using the Genome-to-Genome Distance Calculation (GGDC 2.0) [42] and the ChunLab’s online ANI Calculator (www.ezbiocloud.net/tools/ani) [43], respectively. Genome mining analysis was performed with antiSMASH (version 4.0, Blin K, Oxford, UK) [44].

### 2.6. Activity Evaluation of Strain LD120^T^ against R. solanacearum In Vitro

The phytopathogenic *R. solanacearum* strain FJAT-91 used in this study was kindly provided by Fujian Academy of Agricultural Sciences. *R. solanacearum* was cultured in sucrose-peptone (SP) broth (sucrose 2%, peptone 0.5%, KH_2_PO_4_ 0.05%, MgSO_4_·7H_2_O 0.025%, pH 7.0) for 12 h at 37 °C. Then, 100 μL of bacterial suspension (10^8^ CFU/mL) was plated onto sucrose-peptone agar (SPA) media. A fresh mycelial agar plug (5 mm diameter) of strain LD120^T^, which was cut from the margin of the mycelium grown on the ISP3 medium, was placed in the center of the SPA plate and incubated at 37 °C. The diameter of the inhibition zone was measured after 12 h. Experiments were performed in triplicate.

### 2.7. The Biocontrol Efficacy of Strain LD120^T^ against R. solanacearum

The preventive experiment was performed in this study to evaluate the biocontrol efficacy of strain LD120^T^ against *R. solanacearum* on tomato under greenhouse conditions. The spore suspension of strain LD120^T^ was irrigated in the soil at concentrations of 10^5^ CFU/g, 10^6^ CFU/g, 10^7^ CFU/g, and 10^8^ CFU/g soil, respectively. Tomato seedlings (Maofen 802) were purchased from Shouguang AOLIDE Agricultural Technology Co., Ltd. (Shouguang, China). When plants grew at the four-leaf stage, tomato seedlings were transferred into the pots (one tomato seedling per pot) that contained various concentrations of spores of strain LD120^T^. After growing for a week, 10 mL of cell suspension of *R. solanacearum* (10^8^ CFU/mL) was poured into the soil around the plants. During the experiment, (1) no microbial suspension was added to the soil, “CK”; (2) both the spore suspension of strain LD120^T^ and cell suspension of *R. solanacearum* were added to the soil; and (3) cell suspension of *R. solanacearum* were added to the soil “CKR”. A total of 15 pots were used for each treatment, and the experiment was conducted three times, independently. The pots were kept at 30 °C and 70–80% humidity for 15 days. The disease severity was rated using the five class scale: (0), no symptoms; (1), one leaf partially wilted; (2), two or three leaves wilted; (3), all leaves wilted except the top two or three leaves; (4), all leaves wilted; (5), plant is dead [10]. The disease index (DI) was calculated using the formula: (Σ(plant numbers with the same rating of disease severity × disease rating)/(maximum rating value × total number of plants))× 100. The biological control efficiency was calculated using the formula: ((DI of CKR − DI of the treatment)/ DI of CKR) × 100%.

## 3. Results and Discussion

### 3.1. Polyphasic Taxonomic Characterization of Strain LD120^T^

Identification using the EzTaxon-e server revealed that strain LD120^T^ belonged to the genus *Streptomyces*, with the highest 16S rRNA gene sequence similarity to *S. azureus* NRRL B-2655^T^ (98.97%). Additionally, 16S rRNA gene sequence similarities between strain LD120^T^ and other species of the genus *Streptomyces* were lower than 98.7%. A phylogenetic tree based on 16S rRNA gene sequences showed that strain LD120^T^ formed a phyletic line with *S. anandii* NRRL B-3590^T^ (98.62%) in the neighbor-joining tree (Figure 1); a relationship also recovered in the maximum-likelihood tree (Appendix A). Further, phylogenetic analysis, based on the five housekeeping genes (Appendix A) and whole-genome sequences, showed that strain LD120^T^ formed a separate clade (Figure 2, Appendix A). Based on the phylogenetic trees and 16S rRNA gene similarities, the isolate was thought to be mostly related to *S. azureus* NRRL B-2655^T^ and *S. anandii* NRRL B-3590^T^. dDDH indicated that DNA–DNA relatedness between strain LD120^T^ and *S. azureus* NRRL B-2655^T^ and *S. anandii* NRRL B-3590^T^ were 21.0–25.7% and 25.3–30.1%, respectively, which are much lower than the cut-off point of 70% recommended for the assignment of bacteria strains to the same genomic species [45]. ANI values between LD120^T^ and the two type strains were 78.9% and 83.4%, respectively, whose values were also below the recommended threshold for species delineation (95–96%) [46]. In addition, The MLSA distances between the isolate and the two type strains were 0.108 and 0.118, respectively (Appendix A), which was well above the species level threshold of 0.007, considered to be the threshold for species determination [47].

Morphological observation of a 2-week-old culture of strain LD120^T^ grown on ISP3 medium revealed showed that it had the typical characteristics of the genus *Streptomyces* [48]. Aerial and substrate mycelium were well developed without fragmentation. Long spore chains with warty surfaced spores (0.5–0.6 × 0.7–0.8 μm) were borne on the aerial mycelium (Figure 3). Strain LD120^T^ grew well on ISP2, ISP3, ISP4, ISP6, ISP7, and MBA media but moderately on ISP1, ISP5, NA, and CA media. The colors of aerial mycelium were white, and those of the substrate mycelium varied from pale yellowish to light olive brown (Appendix A). Diffusible pigments were observed on ISP2, ISP4, ISP5, ISP7, NA, and MBA. The significant color variations of aerial and substrate mycelium and the production of diffusible pigments on different media are listed in Appendix A, which shows morphological differences between strain LD120^T^ and the two closely related strains. Strain LD120^T^ could grow at a temperature range of 15–40 °C, but not 45 °C, which can distinguish it from its closely related strains. Hydrolysis of starch and production of urease, as well as utilization of L-arginine and L-glutamic acid, could differentiate the isolate from *S. azureus* NRRL B-2655^T^. Meanwhile liquefaction of gelatin, coagulation of milk, and utilization of carbon sources could distinguish the isolate from *S. anandii* NRRL B-3590^T^. Other details of physiological and biochemical characteristics of strain LD120^T^ compared with its closely related strains are listed in Table 1.

Chemotaxonomic analyses revealed that strain LD120^T^ exhibited characteristics that are typical of members of the genus *Streptomyces* [48]. It contained LL-diaminopimelic acid as the cell-wall diamino acid. The whole-cell sugars were arabinose, glucose, rhamnose, and ribose. The polar lipid profile consisted of diphosphatidylglycerol (DPG), phosphatidylethanolamine (PE), and phosphatidylinositol (PI) (Appendix A). The menaquinones detected were MK-9(H_4_) (42.3%), MK-9(H_6_) (34.3%), MK-9(H_8_) (14.4%), and MK-9(H_2_) (9.0%). The major cellular fatty acids (>10%) were iso-C_16:0_ (22.9%), iso-C_17:0_ (16.9%), anteiso-C_15:0_ (14.2%), and C_16:1_ ω7c (12.7%) (Table 2). The fatty acid profile of strain LD120^T^ was evidently different from those of *S. azureus* NRRL B-2655^T^ and *S. anandii* NRRL B-3590^T^, such as the presence of C_14:0_ in *S. anandii* NRRL B-3590^T^ and a higher amount of anteiso-C_17:0_ in *S. azureus* NRRL B-2655^T^, while there was a higher amount of iso-C_17:0_ in strain LD120^T^.

Therefore, it is evident from the genotypic and phenotypic data that strain LD120^T^ represents a novel species of the genus *Streptomyces*, for which the name *S. physcomitrii* sp. nov. is proposed.

### 3.2. Description of S. physcomitrii sp. nov.

*S. physcomitrii (phys.co.mi’tri.i. N.L. gen. n physcomitrii of the moss Physcomitrium*). Aerobic, Gram-stain-positive actinomycete that forms extensively-branched substrate mycelium and aerial hyphae. Linear spore chains are composed of non-motile spores with a warty surface. Growth occurs at pH 6–11 (optimum pH 7), at 15–40 °C (optimum 28 °C), and in the presence of 0–8% (*w*/*v*) NaCl. It is positive for hydrolysis of starch, Tweens 40 and 80, negative for hydrolysis of aesculin and Tweens 20, has a reduction of nitrate, liquefaction of gelatin, coagulation of milk, production of urease, decomposition of cellulose, and production of H_2_S. It utilizes D-fructose, D-glucose, lactose, D-mannitol, and D-ribose as sole carbon sources but not L-arabinose, D-galactose, meso-inositol, D-maltose, D-mannose, D-raffinose, L-rhamnose, D-sorbitol, D-sucrose, or D-xylose. It utilizes L-alanine, L-arginine, L-asparagine, L-aspartic acid, L-glutamine, glycine, L-serine, and L-threonine as sole nitrogen sources, but not creatine or L-glutamic acid. Cell walls contain LL-diaminopimelic acid as the diagnostic diamino acid, and the whole-cell sugars are arabinose, glucose, rhamnose, and ribose. The menaquinones detected are MK-9(H_4_), MK-9(H_6_), MK-9(H_8_), and MK-9(H_2_). The polar lipid profile consists of diphosphatidylglycerol (DPG), phosphatidylethanolamine (PE), and phosphatidylinositol (PI). The major cellular fatty acids are iso-C_16:0_, iso-C_17:0_, anteiso-C_15:0_, and C_16:1_ ω7c

The type strain is LD120^T^ (=CCTCC AA 2018049^T^ =DSM 110638^T^), isolated from moss (*P. sphaericum* (Ludw) Fuernr) collected in Kunming, Yunnan Province, southwest China. The DNA G + C content of the draft genome sequence of the type strain indicates a value of 73.1 mol% for the species. The GenBank accession number for the 16S rRNA gene sequence and the draft genome sequence of the type strain are MH715906 and JAAWWP000000000, respectively.

### 3.3. Antibacterial Activity of Strain LD120^T^ against R. solanacearum

Strain LD120^T^ showed antagonistic potential against *R. solanacearum* by forming characteristic clear zone (diameter of 12.3 mm), inhibiting the growth of the pathogen in vitro in antagonism assay (Figure 4A). To determine whether the isolate had biocontrol potential against *R. solanacearum* in planta, we evaluated the efficacy of the isolate for control of *R. solanacearum* on tomato seedlings. After 15 days of the inoculation of *R. solanacearum*, some visual wilt symptoms of tomato seedlings were exhibited in both CKR and the treatment groups. In comparison with CKR, the application of strain LD120^T^ significantly reduced the disease severity of bacterial wilt on tomato seedlings (Figure 4B). The spore concentration of 10^7^ CFU/g in soil showed the highest biocontrol efficacy (63.6%) (Figure 4C). This result suggest that strain LD120^T^ has the potential to be developed as a biofertilizer to prevent bacterial wilt on tomato. To mine the biosynthetic potential of strain LD120^T^ associated with antibacterial activity, the draft genome was analyzed using antiSMASH, leading to identification of 37 putative gene clusters, among which, two gene clusters showed 75% and 93% similarities with the biosynthetic gene clusters of salinomycin and fluostatin, respectively, which have been reported to possess significant antibacterial activity [49,50]. However, considering the poor quality of the genome sequence, with a large number of gaps, it may not be related to the antibacterial active components identified with antibiotics and secondary metabolite analysis shell–antiSMASH. Further research is needed to identify the active products in strain LD120^T^.

## 4. Conclusions

An endophytic actinomycete, strain LD120^T^, was isolated from moss (*P. sphaericum* (Ludw) Fuernr). Morphological and chemotaxonomic features, together with phylogenetic analysis, suggested that strain LD120^T^ belonged to the genus *Streptomyces*. Phenotypic characteristics combined with ANI and dDDH values clearly revealed that strain LD120^T^ was differentiated from its closely related strains. Based on the polyphasic taxonomic analysis, it is suggested that strain LD120^T^ represents a novel species of the genus *Streptomyces*, for which the name *S. physcomitrii* sp. nov. is proposed. The type strain was LD120^T^ (=CCTCC AA 2018049^T^ =DSM 110638^T^). In addition, the antibacterial activity of strain LD120^T^ against *R. solanacearum* in vitro and in vivo was assessed, suggesting that it has the potential as a biocontrol agent for controlling bacterial wilt on tomato.

## Figures and Tables

**Figure 1 microorganisms-08-02025-f001:**
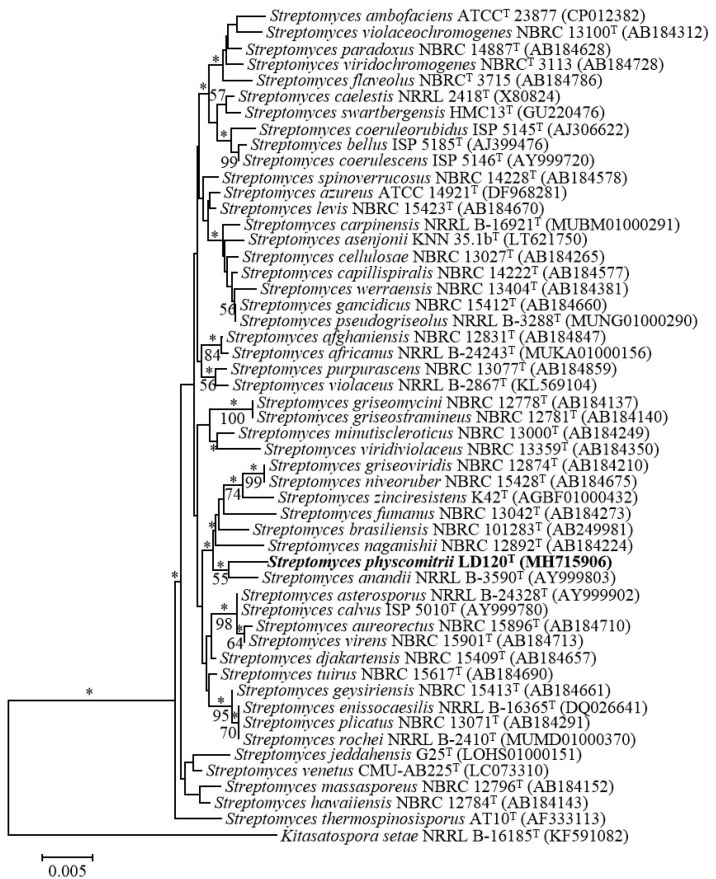
Neighbor-joining tree showing the phylogenetic position of strain LD120^T^ and related taxa based on 16S rRNA gene sequences. Asterisks (*) indicate branches that were also found using the maximum-likelihood method. Numbers at branch points indicate bootstrap percentages (based on 1000 replicates); only values >50% are indicated. Bar, 0.005 substitutions per nucleotide position.

**Figure 2 microorganisms-08-02025-f002:**
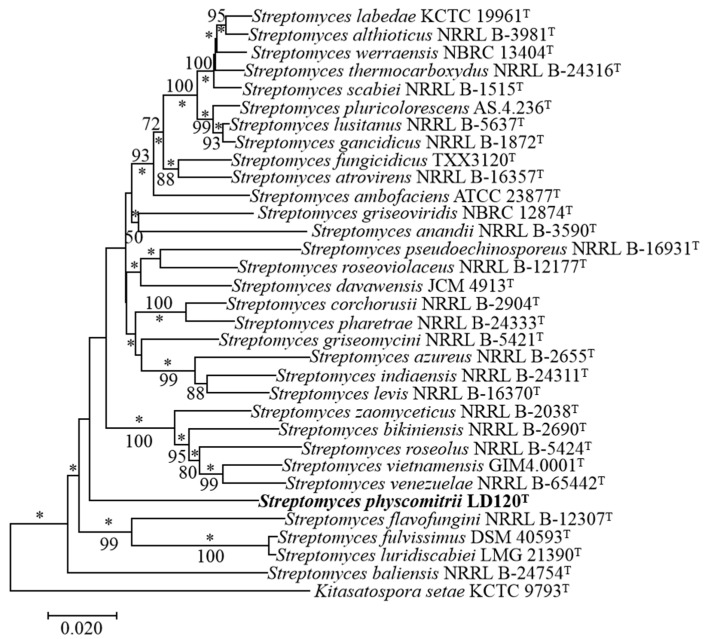
Neighbor-joining tree based on MLSA analysis of the concatenated partial sequences (2481 bp) from five housekeeping genes (*atpD*, *gyrB*, *recA*, *rpoB*, and *trpB*) of strain LD120^T^ and related taxa. Only bootstrap values above 50% (percentages of 1000 replications) are indicated. Asterisks (*) indicate branches also recovered in the maximum-likelihood tree. Bar, 0.02 nucleotide substitutions per site.

**Figure 3 microorganisms-08-02025-f003:**
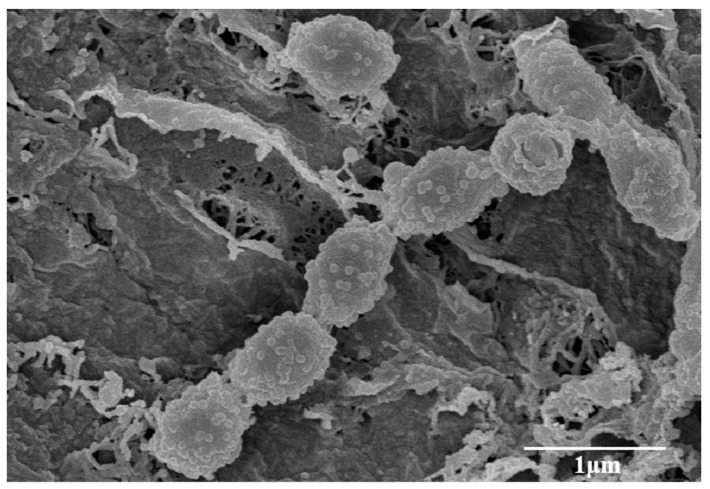
Scanning electron micrograph of strain LD120^T^ grown on International *Streptomyces* Project (ISP) medium 3 agar for 4 weeks at 28 °C. Bar 1 µm.

**Figure 4 microorganisms-08-02025-f004:**
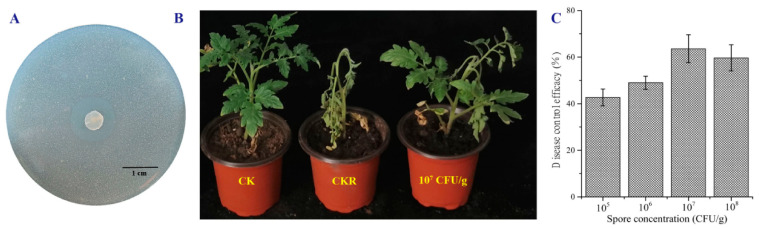
Antibacterial activity of strain LD120^T^ against *R. solanacearum*. (**A**) the antagonistic activity against *R. solanacearum* in the sucrose-peptone agar (SPA) plate. (**B**,**C**) biocontrol assay of inoculation with *R. solanacearum* and strain LD120^T^ on tomato seedlings in the greenhouse.

**Table 1 microorganisms-08-02025-t001:** Differential characteristics of strain LD120^T^ and its closely related strains.

Characteristic	1	2	3
Growth at 45 °C	−	+	+
NaCl tolerance range (*w*/*v*, %)	0–8	0–7	0–10
Liquefaction of gelatin	−	−	+
Coagulation of milk	−	−	+
Hydrolysis of starch	+	−	+
Production of urease	−	+	−
Carbon source utilization			
L-arabinose	−	−	+
D-galactose	−	−	+
*Meso*-inositol	−	−	+
D-maltose	−	−	+
D-sorbitol	−	−	+
D-xylose	−	−	+
Nitrogen source utilization			
L-arginine	+	−	+
L-glutamic acid	−	+	−

Strains: 1, LD120^T^; 2, *S. azureus* NRRL B-2655^T^; 3, *S. anandii* NRRL B-3590^T^. All data are from this study. +, positive; −, negative.

**Table 2 microorganisms-08-02025-t002:** The cellular fatty acid compositions of strain LD120^T^ and its closely related strains.

Fatty Acid	1	2	3
Saturated fatty acids			
C_14:0_	−	−	1.9
C_15:0_	1.0	1.8	−
Unsaturated fatty acids			
C_16:1_ ω7c	12.7	12.8	13.3
C_17:1_ ω7c	9.6	5.4	12.1
Branched fatty acids			
C_17:0_ cycle	5.5	2.2	5.9
iso-C_14:0_	3.0	4.3	6.7
iso-C_15:0_	9.1	11.6	7.5
anteiso-C_15:0_	14.2	13.2	13.9
iso-C_16:0_	22.9	20.1	21.6
iso-C_17:0_	16.9	8.5	9.1
anteiso-C_17:0_	5.2	19.3	7.9

Strains: 1, LD120^T^; 2, *S. azureus* NRRL B-2655^T^; 3, *S. anandii* NRRL B-3590^T^. Values are percentages of total fatty acids. Fatty acids representing <1% in all strains were omitted. All data are from this study. −, not detected.

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
