# Peer review of "Characterization of a Novel Endophytic Actinomycete, Streptomyces physcomitrii sp. nov., and Its Biocontrol Potential Against Ralstonia solanacearum on Tomato"

_microorganisms, 2020, doi:10.3390/microorganisms8122025_

Round 1

Reviewer 1 Report

Dear Authors,

I believe your study is of high relevance for Microorganisms readers. Actinomycetes have been understudied and I think your study is good begining to shed the light on this group of microorganisms. The strategy is nicely conducted and the reults are sound. Unfortunately I think that relevant complementary experiments are highly needed and I could not advice acceptance before you perform them. It is higly needed that you perform preventive and curative experiments in your study. These informations are highly needed to see wether your actinobacteria cans be used as a preventive or curative treatment. Another aspect is related to the use of your actinobacteriabacteria alone on tomato plant and this will be to answer the question wether it can be used as a biofertiliser.

Being very active in this field and having many papers on the same field I can only suggest major revision of your paper that I really woulsd like to see it published after the experiments I suggested. Please consider your paper will once published will be highly relevant for the community and will serve as the very first examples that document use of actinobacteria in plant protection experiments. 

I am looking towards reviewing an improved version of your manuscript that will be higly relevant for microorganisms readers. I find the paper in its current form does not meet microorganisms readers. Microorganisms is now a high standard in microbiology and we all (journal readers and community) would like sound data published.

Please perform the requeired experimenst and proceed as soon as possible to resubmit an improved version that I will be pleased to review.

Congratulations for this nice work and best regards from a scientific colleague

Author Response

In the experiment, we did the preventive experiment of bacterial wilt of tomato by strain LD120T. Tomato seedlings at four leaf stage were transplanted into pots with different spore concentrations. After growing for one week, the pathogen of Ralstonia solanacearum was inoculated to observe the control effect. Because the resistant strain in the experiment is actinobacteria, it needs to take a period of time to colonize in the root after being put into the soil, so as to better control the disease. This experiment is to prevent bacterial wilt.

In order to evaluate whether strain LD120T can affect the growth of tomato, it was also used lone on tomato plant at the concentrations of 105 CFU/g, 106 CFU/g, 107 CFU/g and 108 CFU/g. The results showed that tomato plants with spore suspension had no significant difference from CK, so no photographic records were taken. Strain LD120T had no effect on healthy tomato, so it can be used as a biofertiliser.

Reviewer 2 Report

I send my best compliments to the authors. The manuscript describes a very important research in light of the current needs of the agri-food, environmental and forestry sectors.
In the near future we will see more and more microorganisms used in the fight against plant crop pathogens. However, more and more quality scientific commitment will be needed

Author Response

Thank you very much for your review.

Round 2

Reviewer 1 Report

Dear Authors,

Thanks for the reply. I feel your paper is now ready for publication.

Best regards

Author Response

Thank you very much.